# Zebrafish Models for the Safety and Therapeutic Testing of Nanoparticles with a Focus on Macrophages

**DOI:** 10.3390/nano11071784

**Published:** 2021-07-09

**Authors:** Alba Pensado-López, Juan Fernández-Rey, Pedro Reimunde, José Crecente-Campo, Laura Sánchez, Fernando Torres Andón

**Affiliations:** 1Department of Zoology, Genetics and Physical Anthropology, Campus de Lugo, Universidade de Santiago de Compostela, 27002 Lugo, Spain; alba.pensado.lopez@rai.usc.es (A.P.-L.); juanmanuel.fernandez.rey@rai.usc.es (J.F.-R.); 2Center for Research in Molecular Medicine & Chronic Diseases (CIMUS), Campus Vida, Universidade de Santiago de Compostela, 15706 Santiago de Compostela, Spain; jose.crecente@usc.es; 3Department of Physiotherapy, Medicine and Biomedical Sciences, Universidade da Coruña, Campus de Oza, 15006 A Coruña, Spain; pedro.reimunde@udc.es; 4Department of Neurosurgery, Hospital Universitario Lucus Augusti, 27003 Lugo, Spain

**Keywords:** zebrafish, nanomaterial, nanoparticle, drug delivery, macrophage, immune system, innate immunity

## Abstract

New nanoparticles and biomaterials are increasingly being used in biomedical research for drug delivery, diagnostic applications, or vaccines, and they are also present in numerous commercial products, in the environment and workplaces. Thus, the evaluation of the safety and possible therapeutic application of these nanomaterials has become of foremost importance for the proper progress of nanotechnology. Due to economical and ethical issues, in vitro and in vivo methods are encouraged for the testing of new compounds and/or nanoparticles, however in vivo models are still needed. In this scenario, zebrafish (*Danio rerio*) has demonstrated potential for toxicological and pharmacological screenings. Zebrafish presents an innate immune system, from early developmental stages, with conserved macrophage phenotypes and functions with respect to humans. This fact, combined with the transparency of zebrafish, the availability of models with fluorescently labelled macrophages, as well as a broad variety of disease models offers great possibilities for the testing of new nanoparticles. Thus, with a particular focus on macrophage–nanoparticle interaction in vivo, here, we review the studies using zebrafish for toxicological and biodistribution testing of nanoparticles, and also the possibilities for their preclinical evaluation in various diseases, including cancer and autoimmune, neuroinflammatory, and infectious diseases.

## 1. Introduction

Recent advances in nanotechnology offer the possibility to engineer a wide variety of new nanoparticles and biomaterials with potential application in medicine, but also with commercial interest, providing solutions for numerous sectors in society (i.e., electricity, cosmetics, food packaging, etc.) [1,2]. There is considerable evidence that nano- and microscale materials have unique biological interactions when compared with molecules or bulk materials [3]. Thus, safety assessment results are, of course, of paramount importance [4]. In the case of medical applications, nanomaterials are designed for drug delivery, imaging, diagnosis, sensing, and/or therapeutic purposes. Thus, the benefit/risk for the use of nanotechnologies in this case acquires a different dimension similar to the pharmacological/toxicological profile of other drugs [5]. For these studies, in silico and in vitro methods, including cell- and organ-based assays, are encouraged. However, animal tests are still needed [4,6]. 

In this context, the zebrafish (*Danio rerio*) has become a well-established model for the toxicological and pharmacological screening of new drugs [7] and nanomaterials [8]. Rapid embryo development, small size and transparency, genetic and physiological conservation, ethical and economic advantages have made zebrafish stand out from all other in vivo models [9]. Furthermore, the innate branch of the immune system, including macrophage functions, is well-conserved between humans and the zebrafish, presenting a powerful model for the study of immunotoxicity and also diseases causing and/or caused by inflammatory disorders (i.e., cancer, autoimmune diseases or infectious diseases) [10,11]. Several zebrafish models with fluorescently labeled macrophages have been developed and optimized for studies with a particular focus on the role of macrophages in the toxicological or therapeutic effects of new compounds and nanomaterials. Thus, it is our purpose in this review to collect the information available related to the safety and biomedical testing of new nanomaterials using zebrafish models, with a particular focus on their interaction with macrophages, starting with toxicological and/or biodistribution experimentation, followed by preclinical testing of nanoparticles for therapeutic purposes.

## 2. Zebrafish Innate Immune System: Role of Macrophages

The zebrafish immune system is able to develop both innate and adaptive responses. Despite differences in anatomical sites and time points with respect to mammals. Key cell types, molecular pathways, genetic programs, and transcription factors are highly conserved [12]. Whereas the innate immune system components are already present in early zebrafish embryo stages, the development of adaptive responses does not occur until 2–4 weeks post-fertilization (wpf), and complete immunocompetence is not achieved until 4–6 wpf [13]. Zebrafish innate immune system is composed of similar types of cells than mammals, being macrophages the first leukocytes in development and key players in directing the host immune response [14]. The origin of primitive macrophages in zebrafish occurs at 14–15 h post-fertilization (hpf). These cells migrate to the yolk sac, differentiate, and enter the circulation from 25 hpf [15,16]. The caudal hematopoietic tissue (CHT) at two days post-fertilization (dpf), equivalent to fetal liver or placenta in mammals, acts as a transient hematopoietic site and is a source of embryonic macrophages and neutrophils [17,18]. The thymus produces mature T-cells and the kidney, functional ortholog of mammalian bone marrow, produces myeloid, erythroid, thromboid, and lymphoid cells throughout adulthood [19,20].

Macrophages are phagocytic cells from the innate immune system which can display a wide variety of functions due to their plasticity, versatility and continuous adaptation and response to specific stimuli [21]. In mammals, macrophages have been classically classified according to their polarization extremes observed in the initiation or in the resolution of inflammatory processes [22]. And these cells have been respectively denominated as M1, classically activated or pro-inflammatory macrophages, versus the M2, alternatively activated or wound-healing macrophages [23,24] (Figure 1). These populations have been characterized in terms of gene expression, the pattern of surface molecules, and the production of biological mediators and metabolites [25,26]. An imbalance between M1 and M2 macrophages has been found in pathological tissues and correlated with a worse prognosis of the disease (i.e., high numbers of M1-like macrophages in arthritis or infiltration of M2-like macrophages in solid tumors) [21,27]. Furthermore, taking into account this knowledge, the reprogramming of M1-like macrophages towards M2-like anti-inflammatory effectors appears to be a reasonable strategy for the treatment of some autoimmune diseases [23], while the stimulation of M2-like immunosuppressive macrophages towards M1-like pro-inflammatory-anti-tumor effector cells is a promising approach for the treatment of cancer [21,25,28,29,30,31].

In 2015, Nguyen-Chi et al. provided a seminal report on the polarization of macrophages in zebrafish, showing a great similarity of the M1-like and M2-like phenotypes with respect to mammals [32]. They used the Tg(*mpeg1*:mCherryF) transgenic zebrafish line, which enables to track macrophages [33,34], and generated the Tg (*tnfa*:eGFP-F) line to label M1-macrophages expressing tumor necrosis factor-α (TNF-α). Following fin wounding-induced inflammation or *Escherichia coli* inoculation in zebrafish larvae, they observed a recruitment of macrophages to the wound after amputation or to the muscle after bacteria-inoculation. To characterize the M1/M2 polarization, they mated both lines to generate the Tg(*tnfa*:eGFP-F/*mpeg1*:mCherryF), sorted mCherry+/eGFP+ and mCherry+ cells during the early and late phases of inflammation, and compared their RNA expression patterns, observing high levels of *tnfb*, *il1b* and *il6* pro-inflammatory markers of M1-like macrophages in double-labeled cells, while high levels of *tgfb1*, *ccr2* and *cxcr4b* anti-inflammatory markers were found in the M2-like macrophages only labeled with mCherry [24,32]. Similar results were obtained by Sanderson et al. using the Tg(*irg1*:EGFP)/Tg(*mpeg1*:mCherry) transgenic line, in which irg1 specifically labeled M1-like macrophages upon LPS inoculation or M2-like macrophages upon injection of human metastatic breast cancer cells [35]. Other zebrafish-macrophage-fluorescent reporter lines, such as Tg(*mpeg1*:eGFP)^gl22^ [33] or Tg(*mpeg1*:mCherry)^UMSF001^ [36], have enabled to track the precise behavior and physiology of these cells in vivo, during inflammatory, cancer or infectious processes, thus providing a highly valuable tool to study their role in the context of these pathologies, and also for the screening of new macrophage-targeted therapeutic approaches, including nanoparticles.

## 3. Toxicological and Biodistribution Evaluation of Nanoparticles Using Zebrafish: Focus on Macrophages

### 3.1. Toxicological Studies

The zebrafish is commonly used as an in vivo model for the toxicological evaluation of new compounds and nanomaterials, due to its reduced cost, ease of husbandry, and high fecundity rates [37,38]. Zebrafish and mammals present concordance in toxicological assays ranging from 64% to 100% [39]. Due to its short-life cycle, zebrafish can be used to evaluate intergenerational toxicity by exposing a generation (F0) to a compound and studying the effects on the next generations (F1, F2, etc.) [40,41]. Reproductive impairment can also be studied by assessing sperm motility, egg depositions, and steroid hormone levels [42,43]. 

The usual methods of exposure to new compounds or nanomaterials are microinjection and immersion [44,45]. A widely used harmonized approach is the OECD Fish Embryo Acute Toxicity (FET) Test [46]. The OECD guidelines present immersion exposure as the preferred method and propose that tests should be performed within the first 120 hpf (5 dpf) limit [47,48]. Nishimura et al. reviewed the methodology used in several zebrafish-based developmental toxicity tests and indicated that 5 hpf is the preferred time to start the exposure to a new sample, because at this point embryos are in the late blastula stage and dechorionation can be safely performed [49]. Variations of the FET test have served other researchers to develop assays destined to screen large libraries of compounds or to evaluate organ-specific drug toxicities. For example, Cornet et al. took advantage of the brief zebrafish development to combine in one test the study of cardio-, neuro-, and hepato-toxicity effects in the same individual, enabling recompilation of organ-specific toxicity data for 24 compounds [50]. In a different study, 91 compounds, including pesticides, drugs, and flame retardants, were screened for teratological and behavioral effects, by immersing dechorionated zebrafish embryos in solutions of the different chemicals [51]. In combination with in silico and in vitro assays, zebrafish models were used for the screening of compounds with anti-inflammatory activity from a library of more than 1200 chemicals [52].

Nanotoxicological studies are commonly divided into two fields: environmental health and nanomedicine safety [8]. In both fields, similar types of in vitro and in vivo assays are performed, which in the case of zebrafish models take into account endpoints such as death, hatching rate, developmental malformations, behavioral changes and gene-profiling to assess toxicity [8,53,54]. The main factors influencing nanotoxicological effects, such as composition, particle size, particle shape and surface charge [54], have been investigated in zebrafish [55,56,57]. Zeng et al. studied the effects of Ag nanoparticles (NPs) on zebrafish and found a decreased activity of enzymes implicated in oxidative stress, such as superoxide dismutase, hydrogen peroxide, and malondiadehyde [55]. Another study evaluated how the size of AgNPs could affect toxicity, revealing a higher sensitivity of zebrafish to 20 nm AgNPs versus 100 nm AgNPs [56]. To assess the effects of shape, Abramenko et al. followed OECD guidelines and observed that Ag nanoplates induce higher toxicity than spherical AgNPs [57]. The toxicity of AgNPs on neural development was also evaluated through the study of *gfap* and *ngn1* genetic profiling [58]. The safety of other inorganic based NPs (Au, Mg, Si, Zn) has been studied in zebrafish following similar methodologies [59,60,61,62,63]. Synergistic toxicity of methylmercury, SiNPs and AuNPs including surfactants have also been evaluated in zebrafish, proving it as a useful model to explore the toxicological effects of different compounds in the same organism [64,65]. A more comprehensive toxicological methodology, using zebrafish, and focused on 21 endpoints, was used for the screening of several NPs, leading to the conclusion that surface charge is a major determinant for NP-toxicity [66].

Numerous organic-based nanomaterials have been also tested in zebrafish [67,68,69,70]. The toxicity of polyamidoamine and polypropylenimin dendrimers was tested in zebrafish following the FET method, and with a parallel test in dechorionated embryos, showing for both dendrimers a lower toxicity in chorionated embryos [67]. In a different study, nanographene oxide with an external layer of polyethilenglycol was microinjected in zebrafish to evaluate its toxicity profile and effect on angiogenesis [68]. In another study, Teijeiro-Valiño et al. examined the toxicity of polymeric nanocapsules with an outer shell of hyaluronic acid and protamine [69].

The interaction of NPs with macrophages was investigated using zebrafish models, having important implications in the biocompatibility-toxicity, as well as in the biodistribution of the NPs (reviewed in the next section) [70,71,72,73,74]. On one hand, NPs may trigger direct cell toxicity through different molecular mechanisms [75], on the other hand numerous NPs can also trigger inflammatory responses when they are recognized as foreign agents by immune cells (i.e., macrophages) [5]. In response to NPs, the immune cells commonly produce soluble mediators, such as cytokines, chemokines, and complement factors, which result in the recruitment of more cells and development of acute or chronic inflammatory responses [76]. We should also remember that, in some cases, the inflammatory cells are capable of the biodegradation of NPs, e.g., through oxidative stress or enzymatic degradation (i.e., myeloperoxidase or eosinophil peroxidase) [77]. The balance of these responses is of foremost importance for immunotoxicological studies. As an example, graphene oxide immunotoxicity was evaluated by measuring glutathione, malondialdehyde, superoxide dismutase, catalase, and genetic profiling in adult zebrafish. TNF-α, IL-1β and IL-6 expression levels were significantly increased in a dose-dependent manner [74]. Similarly, it has been reported that several types of metallic NPs, such as gold NPs (AuNPs), silver NPs (AgNPs), or zinc oxide NPS (ZnO-NPs), may induce oxidative stress and disrupt signaling pathways related to innate immune responses [78,79,80]. Other studies involving drugs, nanocarriers, and macrophages practiced their experiments on zebrafish [81,82]. For example, poly(lactic-co-glycolic acid) NPs (PLGA-NPs) loaded with thioridazine were evaluated for toxicity and their therapeutic efficacy was tested in murine macrophages, human macrophages and zebrafish models [81]. These immunotoxicological studies, using zebrafish as in vivo model, demonstrate the relevance of safety assessment for nanomaterials with potential application in a variety of industrial sectors (i.e., cosmetics, paints, food package, etc.), and they also provide pharmacological/toxicological information of interest for their application with medical purposes.

### 3.2. Biodistribution

Nanotechnology is commonly used to improve the biodistribution/pharmacokinetics of drugs. Thus, biodistribution moves from being determined by the drug’s physicochemical characteristics to be dictated by the NP’s-features. Besides, by tuning the physicochemical properties of the NPs, a preferential accumulation in specific organs and/or cell populations can be achieved [83]. Taking this into consideration, zebrafish is gaining importance in the nanomedicine field as a simple and reliable model to screen biodistribution of drugs and/or NPs. The transparency of the embryos [84] and the availability of transgenic lines offer the possibility to study in biodistribution studies the role of the interaction of NPs at the cellular level, for example, by using macrophage-labelled embryos [85]. In this context, different scenarios can happen depending on the aimed cellular and/or molecular target: macrophages can be the final target for the nanomedicine, such as in the case of autoimmune or infectious diseases, and nano-vaccines [86], or they can be a cellular population to be avoided, due to the undesirable clearance of the therapeutic entity mediated by these phagocytic cells, such as in the case of drugs targeted towards cancer cells or others [72]. Of note, in the context of cancer both approaches, targeting and/or avoiding macrophages, could be intended to improve the efficacy of certain drugs, directed towards macrophages (i.e., immunotherapy) or to cancer cells (i.e., chemotherapy), respectively [87].

The overall biodistribution of a NP is governed by the combined effect of the administration route, the dose and the nanostructure’s composition and properties (Figure 2). As previously described for the toxicity, the size, shape, charge and flexibility are also critical parameters that determine the final fate of the NPs [83,88]. The circulation time of NPs administered intravenously (i.v.) depends, mainly, on their interaction with immune and/or endothelial cells [89]. In fact, the majority of the NPs i.v. administered are cleared in the liver, by macrophages or by sinusoidal cells [85,90]. Part of these NPs are also captured by circulating monocytes/macrophages, and/or other immune cells, such as neutrophils, before reaching their target tissue, as it was recently shown for gelatine nanospheres [86] and liposomes [91]. In zebrafish, liposomes of 60 nm showed a decreased macrophage uptake as compared to larger 120 nm liposomes, which were more accumulated in the spleen [91]. Apart from immune cells, uptake by endothelial cells can also be responsible for the limited circulation time of certain NPs. For example, polystyrene (PS) NPs of 1000 nm were found adhered to the endothelium some minutes after the injection. On the contrary, with a lower adherence to the endothelium, 200 nm PS NPs showed a more prolonged circulation time. Our group evaluated the diffusion of NCs of different sizes and surface charges in zebrafish embryos [70]. These NPs are versatile nanosystems with applications in different fields, such as cancer, vaccination, and ocular diseases [92,93,94,95]. Both positively and negatively charged NCs of 70 nm spread faster and at a higher extent than medium size NCs (200 nm) after i.v. injection in zebrafish. Chitosan NCs (positively charged) were found to be attached to the endothelial cells of the blood vessels at a higher degree than inulin NCs (negatively charged). As a whole, zebrafish injected with the 70 nm chitosan NCs showed the highest intensity, due to the fluorescent-NCs, in the circulatory system, and also higher accumulation in other tissues, such as the brain and visceral organs. All NCs showed certain degree of co-localization with GFP-labelled macrophages. However, the positively charged NCs showed the highest accumulation in these cells. In relation to their particle size, 200 nm NCs interacted with macrophages from early time points (0.5 h), whereas 70 nm NCs presented less interaction at 0.5 h, but their accumulation in these cells increased with time. From this and other studies, it can be concluded that larger particles are commonly attached or captured by macrophages and/or endothelial cells, thus showing shorter circulation times.

Particle shape and flexibility have also an impact on NPs biodistribution. Generally elongated and rod-shaped NPs present a prolonged time in circulation compared with spherical particles, especially if they are flexible [96]. This effect was observed in different animal models (i.e., mice or pigs) [97], and also zebrafish [98]. In this case, the non-spherical NPs were poorly taken up by macrophages because of hydrodynamic shearing. However, other authors found that spherical NPs present prolonged circulation time versus NPs with fibrillar morphologies upon i.v. administration in zebrafish [99]. Overall, it is important to highlight that the final biodistribution and toxicological behaviour of a NP is orchestrated by the interplay between its different physicochemical properties (i.e., size, charge, and shape), and focusing only in one parameter can be misleading. 

The composition of the NP’s surface and/or its functionalization with specific molecules is also a key factor affecting its biodistribution. For instance, liposomes presented substantially higher circulation time after i.v. administration in zebrafish than PS NPs of similar size (nm) [72]. The different circulation patterns can be explained by the high affinity of PS NPs for the endothelium, while liposomes showed a slow macrophage-mediated clearance. With respect to decoration of NPs surface, polyethylene glycol (PEG) and other molecules have been used to avoid macrophage interaction, while ligands for specific receptors on macrophages-surface or other cells have been investigated. Functionalization with PEG chains is commonly implemented to “mask” a therapeutic agent from the immune system, reducing immunogenicity and antigenicity. PEGylated nanosystems show longer circulation times after i.v. administration than non-PEGylated ones, by reducing their interaction with opsonins and making NPs “invisible” to the immune system [72,89]. As an example, liposomes commonly cleared from the circulation by macrophages, when PEGylated, increased their circulation time, and their elimination by macrophages, although still occurring, is delayed [89]. In the case of PEGylated PS NPs, the increase in their circulation time has been attributed not only to their reduced capture by macrophages, but also to their reduced interaction with the endothelial cells [72]. The degree of PEGylation and its molecular weight also impacts on NPs biodistribution, showing decreased clearance by macrophages for liposomes decorated with a higher density of PEGs or increased molecular weight [91]. Other polymers with “shielding behaviour” have been used, such as polysarcosine or poly(N,N-dimethylacrylamide), showing an increase in NP-circulation time in zebrafish models [89].

Zebrafish models have also been used to study NP-biodistribution through other routes of administration. Upon intramuscular (i.m.) injection in zebrafish, NPs generally only spread along the muscle tissue close to the injection site, at least for short times [86]. As time goes by, the NPs are commonly internalized by macrophages to be cleared [86]. In our previously mentioned work, after i.m. injection, the 70 nm NCs disseminated further than 200 nm NCs, spreading through all the myomere and limited by myosepta [70]. Positively charged NCs (chitosan based) recruited more macrophages than negatively charged (inulin based) NCs, following a similar pattern to the i.v. route. To study the mucosal administration of new therapies using zebrafish, NPs can be simply incubated in water. Verrier and colleagues found that 200 nm PLA-NPs were able to cross the epithelial barrier of different mucosae (nasal, gills, gut, and skin) and be accumulated in antigen presenting cells, such as dendritic cells, macrophages and B cells in the gills and skin [100]. Interestingly, the NPs were able to enter the bloodstream through the gills, and to then reach internal organs, such as the liver and kidney. Our group showed the importance of surface composition in the ability of NCs to diffuse through the chorion before the zebrafish hatching [69]. The presence of a PEGylated surfactant is supposed to favour the diffusion of these NCs through the thick chorion barrier. Using hatched embryos, we demonstrated that hyaluronic acid NCs are not able to cross the epidermis. However, protamine-hyaluronic acid NCs were internalized and reached the yolk sac, the stomach, the esophagus and the olfactory pit. In this case, protamine, a well-known cell penetrating peptide, could be important to facilitate the internalization and transport through the zebrafish skin epithelium [69]. The particle size also influences de biodistribution by the mucosal routes. For instance, after incubation with zebrafish in water, coumarin nanocrystals of 70 nm showed better permeability across the chorion, blood brain barrier, blood retinal barrier and gastrointestinal barrier than their counterparts of 200 nm. Besides, the smaller nanocrystals accumulated at a higher extent in internal organs via lipid raft-mediated endocytosis [101]. The importance of the particle shape was demonstrated by Vijver and his group [102]. After waterborne exposure of zebrafish embryos to gold NPs of different shapes, the macrophage’s abundance was higher for urchin-shaped NPs compared to the spherical ones.

## 4. Preclinical Testing of Nanoparticles Using Zebrafish Models of Disease: Relevance of Macrophages

### 4.1. Cancer

Due to its unique features, zebrafish tumor models, mainly induced by transgenesis or xenotransplantation, are increasingly being used for cancer research and discovery of new antitumoral drugs [103]. Xenografts are routinely performed by implantation of human or murine cancer cells into different anatomical sites of zebrafish embryos (yolk sac, duct of Cuvier or perivitelline space) to obtain heterotopic or eventually orthotopic in vivo tumor models. The injection of labeled cancer cells in “transparent zebrafish” allows to track their survival, progression, migration, and interaction with the host microenvironment [104,105]. For the transgenic models, genetically modified zebrafish lines, mainly based on the expression of human oncogenes driven by tissue-specific or ubiquitous promoters (e.g., BRAF^V600E^, HRAS^G12V^ or KRAS^G12V^) have been used [106,107,108]. While the transgenic models are commonly preferred for the mechanistic understanding of tumor development and/or interaction of cancer cells with the tumor microenvironment (TME), xenografts are more used for drug screenings. As the generation of stable transgenic zebrafish lines requires several months, it is technically challenging and less cost-effective when compared to the simple injection of tumor cells in xenografts [109]. In addition, xenograft models have been recently used to understand the metabolic and/or stem cell properties of cancer cells and they also offer the possibility to study patient-derived cancer cells in vivo [110,111].

The study of macrophages in tumors is nowadays a very active field of research. Tumors are complex tissues, comprising a heterogeneous population of cells plus the extracellular matrix they produce, which constitute the TME [112,113,114,115]. The cellular fraction is composed of malignant cancer cells, as well as endothelial cells, cancer-associated fibroblasts, and tumor-associated macrophages (TAMs), with the latter being the latest the most abundant cell type [116]. In human tumors, TAMs originate mostly from circulating precursor monocytes, but resident macrophages can be originally present in the tissue, later developing in a tumor [117]. TAM infiltration in tumor tissues has been shown to support tumor growth, angiogenesis, invasion and metastasis, and their high density in tumors has been correlated with tumor progression and resistance to therapies [118]. The secretion of CSF-1, CCL2 and VEGF, by cancer cells, induce the recruitment of macrophages towards the TME, and the Th2 cytokines IL-4, IL-13, IL-10 and TGF-β, metabolic signals (i.e., lactic acid and hypoxia) produced also by Treg and TAMs are key drivers of immunosuppression [116]. With the aim to better mimic human tumors, including the TME, zebrafish xenografts are being continuously improved, using reporter lines for analysis of vasculature [119], neutrophils [120] and/or macrophages [32,33,35]. Orthotopic xenotransplantation, consisting in the injection of tumor cells in the equivalent anatomical site, and the use of patient-derived xenografts (PDX), allows the preservation of tumor cells’ original phenotype, and represents a further step to recapitulate the TME and tumor cell-host interactions [121]. The knowledge and practice acquired using these models will not only be useful for a better understanding of tumors, but also determinant for the development and evaluation of new antitumoral therapies [122], including nanotechnological strategies.

An enormous variety of nano-oncologicals (nanostructures for the treatment of cancer) have been developed and evaluated with the main purpose to improve the delivery of pharmacological molecules to the TME and reduce off-target effects [123,124,125]. NPs have been designed to improve the efficacy of classical chemotherapies, immunomodulatory drugs, but also delicate molecules such as nucleic acids, and even new cellular therapies [126,127]. Main therapeutic strategies to target and impact on TAMs include: (i) inhibition of TAM recruitment to the tumor, (ii) direct killing of TAMs, (iii) re-education of TAM from their M2-like protumoral phenotype into a M1-like antitumoral phenotype [87]; offering promising opportunities to switch the tumor-promoting immune suppressive microenvironment, characteristic of tumors rich in macrophages, to one that kills tumor cells, is anti-angiogenic and promotes adaptive immune responses [87,128]. 

In this context, zebrafish models offer the possibility to test the antitumoral effect of new nano-oncologicals, not only at the level of tumor growth in vivo [109], but also at the cellular level within the tumor (i.e., cancer cells or TAMs in vivo). Relevant examples of nano-oncologicals tested in zebrafish are reviewed below and summarized in Table 1. 

Nadar and colleagues investigated the ability of drug-loaded hydroxyapatite NPs to release active therapeutics in vivo [129]. To this end, they used a zebrafish xenograft model with MDA-MB-231 breast cancer cells to test the antitumoral efficacy of kiteplatin-pyrophosphate-loaded hydroxyapatite NPs (PtPP-HA). Two days after the co-injection of cancer cells and the platinum-loaded HA NPs into 48 hpf zebrafish blood circulation, they found a significant decrease in survival of breast cancer cells [129]. Different types of metal oxide-NPs, not loaded with additional pharmacological molecules, have also been tested for antitumoral efficacy. ZnO-NPs were evaluated using a gingival squamous cell carcinoma xenograft model, showing antitumoral activity via induction of reactive oxygen species (ROS) and reduction of anti-oxidative enzymes with consequent oxidative damage to cells and tissues [130,138,139]. DiL-labeled Ca9-22 cells were implanted into the yolk sac of 48 hpf Tg(*fli1*:EGFP) embryos and fluorescence monitoring showed no effect on survival but dose-dependent inhibition of tumor growth [130]. The physicochemical properties of nanomaterials have also been exploited for photodynamic therapy (PDT) and tested in zebrafish. For instance, Jimenez et al. developed, and tested in zebrafish tumor xenografts, porous porphyrin-based organosilica NPs (PMOsPOR-NH2) [131]. At 2 dpi, treated-embryos were subjected to two-photon-excited photodynamic therapy (TPE-PDT) and a complete extinction of GFP-cancer cells was observed. Additionally, to analyze the efficiency of gene delivery, the same NPs were complexed with anti-GFP siRNA and co-injected together with GFP mRNA at one-cell stage, leading to a reduction in GFP expression a few hours later, thus also revealing the potential use of these NPs for gene therapy. Similarly, small-seized non-porous porphyrin-based bridged silsesquioxane NPs (PORBSNs) functionalized with PEG and mannose (20–30 nm) were developed for i.v. injection and tested in zebrafish with MDA-MB-231-GFP cells subjected to TPE-PDT, showing decreased tumor growth after irradiation [132]. Moreover, with targeting purposes, Peng et al. synthesized fluorescent cellulose acetate NPs functionalized with folate groups for their preferential accumulation in epithelial cancer cells overexpressing folic acid receptors [140]. These NPs can be tuned within the entire UV-VIS-NIR spectrum and were capable to target tumors in vivo. To test the combinatorial antitumoral efficacy of chemotherapy and radiotherapy using NPs, Wu et al. synthesized G4.5 polyamidoamine (PAMAM) dendrimers conjugated with l-cysteine (GC), acting the later as radiosensitizer, and further loaded with doxorubicin (PAMAM-GC/DOX) [133]. Cervical carcinoma HeLa cancer cells labeled with carboxyfluorescein succinimidyl ester (CFSE) were injected into the yolk sac at 2 dpf. Embryos were immersed in PAMAM-GC/DOX and further exposed to γ-radiation at 3 dpf. Synergistic antitumoral effect for the combination chemotherapy and radiotherapy with NPs was confirmed, versus only radiation or free DOX with or without irradiation. Likewise, PAMAM dendrimers, functionalized with PEG using a hypoxia-induced sensitive linker, were loaded with DOX and hypoxia-inducible factor 1a siRNA (PAMAM-DOX-siHIF) [134]. The antitumor activity of free DOX, PAMAM+DOX, and PAMAM-DOX+si-HIF was tested upon implantation of MCF-7-CM-DiL breast cancer cells into the perivitelline space of 48 hpf Tg(*fli1*:EGFP) embryos, and intracardiac injection of free DOX, PAP+DOX and PAP-DOX+si-HIF at 1 dpi, showing the best results for the “triple-therapeutic combination” (PAMAM-DOX+si-HIF). Despite these promising results in zebrafish, we still foresee some limitations for the translation of some results to the clinic, in part due to the differences in zebrafish anatomy versus mammals, because of the need for deeper penetration of radiotherapy in hidden tumors. However, a positive experience was recently reported by Costa et al., showing the utility of zebrafish to distinguish radiosensitive from radioresistant tumors using colorectal cancer cell lines and patient biopsies, and clinical response was correlated with induction of apoptosis in zebrafish [141].

Several redox- and pH-responsive-NPs have been developed, to favor the control delivery of drugs in the reducing and acidic TME, and several zebrafish models were optimized for the testing of these NPs. Transgenic mifepristone-inducible liver tumor zebrafish line expressing the enhanced green fluorescence protein (EGFP)-Kras^v12^ oncogene [142], as a model of hepatocellular carcinoma with elevated glutathione and liver acidity, was used [143,144]. While embryos treated with free DOX died at 2–3 dpi, lower toxicity and sustained regression of tumor size was observed for the DOX-loaded-NPs, demonstrating improved drug release to liver tumor cells and lower systemic toxicity [145]. A pH-sensitive hydrazone-linked DOX nanogel (Nanogel^DOX^) was i.v. injected in zebrafish, previously implanted with B6 mouse melanoma cells into the neural tube at 48 hpf, showing significant reduction in tumor growth versus no effect for free DOX. [135]. Redox-responsive silica-gold nanocomposites functionalized with transferrin and loaded with DOX (Tf-DOX-ReSi-Au) were injected in a zebrafish colorectal cancer model, showing positive antitumoral activity without the typical DOX-related cardiotoxic adverse effects [136]. Others have tested the capacity of fluorescent-NPs to detect cancer cells in vivo, using similar zebrafish tumor models [146,147]. 

To understand the interaction between immune and tumor cells, several zebrafish xenograft models have been developed. These models have allowed to study the role of immune cells in tumor vascularization and invasion [148], the dynamic interaction between immune and cancer cells [149], or the positive correlation between the number of immune cells recruited to the tumor site and the degree of angiogenesis [150,151]. Póvoa et al. showed distinct engraftment profiles from the same patient at different stages of tumor progression in colorectal zebrafish xenograft models and explored the innate immune contribution to this process [152]. While cells derived from the primary tumor were able to recruit macrophages and neutrophils, thus being rapidly cleared (regressors), those cancer cells derived from a lymph node metastasis polarized macrophages towards a M2-like protumoral phenotype, engrafting very efficiently (progressors). Interestingly, mixing both types of cells resulted in decreased regressors clearance, reduced numbers of innate cells and increased M2-like polarization. Furthermore, depletion of macrophages resulted in a significant increase in the engraftment of regressors. These results provide the first experimental evidence of therapeutic manipulation of macrophages in zebrafish tumor models [152]. Detailed studies, using NPs for reprogramming TAMs into antitumoral M1-like macrophages, have still not been performed using zebrafish models of cancer. Nevertheless, a few investigations have explored the circulation time of NPs, their accumulation at the tumor site and their specific uptake by macrophages in tumors using these models. Evensen et al. were the first to image the accumulation of NPs to human tumor-like structures in a zebrafish xenograft model [72]. Comparing labeled polystyrene NPs and liposomes, with or without PEGylation, upon injection into the posterior cardinal vein of 2 dpf embryos, they noted that PEG-liposomes displayed the longest circulation time due to their lower affinity to the endothelium, the lowest macrophage uptake, and the highest survival rate. Using xenografts, with melanoma and kidney cancer cells, the PEG-liposomes showed a specific and rapid accumulation into the tumor and outside the vasculature after only 2–5 h post- injection (hpi) [72]. Kocere and colleagues demonstrated the selective accumulation of Cy5-labelled poly(ethylene glycol)-block-poly(2-(diisopropyl amino) ethyl methacrylate) (PEG-PDPA) NPs in TAMs using a melanoma xenograft model [137]. By injection of B6 mouse melanoma cells (labeled with GFP or RFP) in the neural tube of 3 dpf transgenic embryos ((Tg(*fli1a*:EGFP, Tg(*mpeg1*:mcherry), Tg(*mpx*:GFP)) [33,119,153], they were able to observe tumor growth, angiogenesis, and accumulation of macrophages, but not neutrophils, within the tumor at 10 dpf. Additionally, xenografts in 3 dpf Tg(*mpeg1*:GAL4/UAS:*NTR*-mCherry) embryos, which express a nitroreductase in macrophages in presence of metronidazole causing their selective apoptosis, revealed a slightly increased tumor growth when macrophages were absent. The injection of PEG-PDPA NPs, showed selective accumulation of NPs and increased number of macrophages at the tumor site. Moreover, a small fraction of NPs was internalized by cancer cells and TAMs. These DOX-loaded-PEG-PDPA NPs were i.v. injected in the melanoma model at 1 dpi, showing reduced toxicity, decreased proliferation, and increased apoptosis of cancer cells six days after treatment [137].

As a whole, these studies provide a consistent knowledge and experience for the development of several types of tumors in zebrafish and the testing of nano-oncologicals with different features. Zebrafish models provide excellent opportunities for genetic modifications and for in vivo evaluation/tracking of innate immune cells, being these key aspects for the testing of NPs. A major limitation for the use of zebrafish embryos to test antitumoral therapies is their lack of adaptive immune system. This challenge has partially been addressed through the suppression of the adult immune system, either by γ-irradiation, dexamethasone treatment [154,155,156] or more recently using adult immunocompromised strains [157,158,159], followed by the injection of human or murine immune cells (i.e., T cells). These adult zebrafish xenografts enable a closer resemblance of cell–tumor microenvironment interactions, a longer tumor engraftment and a clinically-relevant dose response [160,161]. Remarkably, Yan et al. created an optically clear, homozygous mutants (*prkdc−/−, il2rga−/−*) which lacks T, B, and natural killer (NK) cells. These animals survived at 37 °C (optimal for mammal cells), robustly engrafted a variety of human tumor cells and subsequently responded to drug treatments. Importantly, similar histological and molecular features in both fish and mouse xenografts were confirmed and pharmacokinetics of antitumoral treatments, such as olaparib and temozolomide, were comparable to that found in both mouse preclinical models and humans. Nevertheless, this adult immunocompromised strain must be improved, as several cancer cell types failed to engraft into the model and pre-treating fish with clodronate liposomes to deplete macrophages was required [159]. Additional disadvantages of zebrafish models are that some strains do not breed, develop gill inflammation, and likely autoimmunity [158], or they could be quite prone to infection and require specialized food and antibiotic treatment [162], thus raising the cost of maintenance. Despite these challenges, the results demonstrate that zebrafish tumor models are important tools with high potential to improve the translation of nano-oncologicals towards the clinic.

### 4.2. Autoimmune Diseases

#### 4.2.1. Inflammatory Bowel Disease

Inflammatory bowel diseases (IBDs) refer to chronic inflammatory disorders of the gastrointestinal tract which comprise both Crohn’s disease and ulcerative colitis. Although their etiology is not clear, they are thought to be a result of host genetic susceptibility and environmental factors (e.g., diet) [163,164], which lead to altered interactions between gut microbiota and the intestinal immune system [165]. Intestinal homeostasis is partly maintained by resident macrophages with enhanced phagocytic and bactericidal activity and decreased production of pro-inflammatory cytokines [166]. Nevertheless, when gut dysbiosis and further disruption of normal mucosal immunity occur, monocytes are continuously recruited to become inflammatory macrophages, they participate in the inflammatory response and contribute to chronic intestine inflammation [167]. In this context, zebrafish models are useful to study the relationship between immune system and inflammation. For instance, Coronado et al. treated fish with a previously established inflammatory diet [168] and found a strong increase in the number of neutrophils, macrophages and T helper cells recruited to the gut [169]. Looking into the genetic susceptibility, Kaya et al. were able to validate the implication of GPR35-expressing macrophages in intestinal immune homeostasis and inflammation by generating a zebrafish mutant line [170]. Other researchers have shown the potential of zebrafish for the screening of drugs to treat IBDs [171,172,173]. With regard to NPs, to date, we only found one study where the administration of copper NPs to zebrafish resulted in intestinal developmental defects, through ER stress and ROS generation, showing similar alterations to IBD patients (Table 2) [174]. Thus, these studies provide the starting point for the study of IBD pathology and testing of NPs which might offer new solutions to patients suffering these intestinal disorders.

#### 4.2.2. Type I Diabetes Mellitus

Type 1 diabetes mellitus (T1DM) is an autoimmune disease caused by immune-mediated progressive destruction of the pancreatic β-cells, driven by the interaction of multiple environmental and genetic factors. The pathogenesis of T1DM is characterized by the infiltration of islet antigen-specific T cells and pro-inflammatory APCs associated with impairment of Foxp3+ Tregs. The destruction of β-cells leads to the loss of ability to produce insulin and in turn, to chronic hyperglycemia [175,176]. T1DM treatment is mainly based on lifelong insulin replacement therapy and several nanoparticles have been designed to improve its administration [177,178]. Others have explored the inhibition of the destructive autoimmune response against insulin-producing β-cells, for example by regulating T-cell autoreactivity as a therapeutic approach [179,180]. In zebrafish models, to mimic T1DM, the destruction of β-cells has been achieved by either surgery [181], chemical destruction [182], or genetic ablation [183], followed by their subsequent regeneration ability allowing to study the mechanisms of β-cells regeneration and also the testing of antidiabetic drugs [184,185]. A number of chemical screens to induce β-cell generation in zebrafish have been reported [186,187,188] and the antidiabetic effect of different bioactive molecules and NPs has been tested (Table 2).

Silver nanoparticles loaded with *Eysenhardtia polystachya* (EP/AgNPs), with a spherical shape and diameter of 5–21 nm, were tested on glucose-induced diabetic adult zebrafish and the results confirmed the effectiveness of NPs in ameliorating hyperglycemia [189]. The utility of quercetin NPs (NQs) in ameliorating diabetic retinopathy, a common complication derived from diabetes was shown by Wang et al. [190]. Chemically induced diabetes and diabetic retinopathy were established in adult zebrafish, and further treatment with NQs led to a reduction in glucose blood levels as wells as to the improvement of different morphological, behavioral, and biochemical parameters linked to diabetic retinopathy [190]. Others have evaluated the biocompatibility/toxicological profile of different types of NPs with potential antidiabetic activity, such as peptide-major histocompatibility complexes-NPs or curcumin encapsulated in polycaprolactone-grafted oligocarrageenan nanomicelles [191,192]. These studies provide a solid basis for the further use of zebrafish models to screen new antidiabetic nanotechnological approaches.

#### 4.2.3. Rheumatoid Arthritis

Rheumatoid arthritis (RA) is a chronic inflammatory and autoimmune disease characterized by synovial and joint swelling, pain, and bone destruction [193]. The pathogenesis of RA is a multistep process, initially starting outside the joints by the aberrant activation of antigen-presenting cells, triggered by genetic or environmental causes, which leads to the activation of a pro-inflammatory cascade, production of autoantibodies as well as altered T-cell and B-cell cross-activation [194,195]. These events lead to monocyte recruitment to the diseased tissue and activation and polarization of macrophages towards a M1-like pro-inflammatory phenotype, boosting the inflammatory cascade [196]. Current RA-treatments include nonsteroidal anti-inflammatory drugs (NSAIDs), corticosteroids, disease-modifying antirheumatic drugs, and different natural substances to reduce joint inflammation [197]. With the aim to mimic human disease, improve its understanding and test new therapies, several animal models have been developed [198,199], and zebrafish models could be also implemented as a useful tool. Zebrafish have served as a model to evaluate the anti-inflammatory properties of different synthetic and natural compounds, although no reports on nanomaterials are available yet. *Clerodendrum cyrtophyllum* Turcz is a commonly used plant in Vietnam for treating RA. Ngyuyen et al. confirmed the anti-inflammatory properties of ethanol extracts of this plant in a copper-induced inflammation zebrafish model, via downregulation of inflammation mediators and pro-inflammatory cytokines (e.g., *cox-2* or *il-1**β*) [200]. Similarly, Jiang et al. demonstrated anti-inflammatory effects of isothiocyanate prodrugs in a zebrafish neutrophilic inflammation model, as the number of migrating neutrophils in treated zebrafish was smaller than in the control group [201]. Wang et al. demonstrated the inhibition of cyclooxygenase, as the anti-inflammatory mechanisms of action of *Gentiana dahurica* roots, in a zebrafish model of induced production of cyclooxygenases 1 and 2 [202]. Genetic approaches using zebrafish were applied to study the role of *c5orf30*, whose variants have been associated with RA. Upon tail transection in the *c5orf30* knockdown fish an increased recruitment of macrophages to the wound site was observed, confirming the anti-inflammatory role of *c5orf30*, with implications in RA [203]. With reference to early diagnosis, Feng et al. designed and synthesized fluorescent probes for the quantitative detection of hypochlorous acid, a biomarker of RA, and confirmed their efficiency in an LPS-induced inflammatory model of adult zebrafish [204]. Additionally, zebrafish has served as a platform to evaluate the toxicity of new treatments with potential application in RA [205,206]. Although mammalian models of RA are needed before the clinical translation of new therapeutic approaches, these zebrafish models provide a valuable tool for the initial screening of innovative nanotechnological approaches to treat RA.

#### 4.2.4. Neuroinflammatory and Neurodegenerative Diseases

Several zebrafish models have been established and optimized for the understanding of neuroinflammatory or neurodegenerative diseases [207,208]. In parallel, different types of NPs have been engineered for the treatment and/or diagnosis of neuro-related disorders [209,210]. Interestingly, zebrafish models have also been used to study the neurotoxic or neuroprotective effects of a wide variety of NPs and biomaterials [207]. Below, we provide some examples of investigations using zebrafish models and NPs to improve drug targeting and/or efficacy in the context of neuroinflammatory and neurodegenerative diseases (Table 2).

Neuroprotective effects of NPs in Parkinson’s disease (PD) using zebrafish models have been observed, commonly mediated by the antioxidant and/or neuro-antiinflammatory activity of these NPs. Bacopa monnieri platinum NPs (BmE-PtNPs) demonstrated the same activity of Complex I, as that of oxidizing NADH to NAD(+), suggesting that BmE-PtNPs could be a potential medicinal substance for oxidative stress mediated disease with suppressed mitochondrial complex I as it happens in PD. Hence, in 1-methyl-4-phenyl-1,2,3,6-tetrahydropyridine (MPTP)-induced experimental Parkinsonism in zebrafish model, BmE-PtNPs pretreatment significantly reversed the toxic effects of MPTP by increasing the levels of dopamine, its metabolites, GSH, and activities of GPx, catalase, SOD and complex I, and reducing levels of MDA along with enhanced locomotor activity [211]. Schisantherin A (SA) is a promising anti-Parkinsonism Chinese herbal medicine and SA nanocrystals (SA-NC) were used to reverse the MPTP-induced dopaminergic neuronal loss and locomotion deficiency in zebrafish. This strong neuroprotective effect of SA-NC may be partially mediated by the activation of the protein kinase B (Akt)/glycogen synthase kinase-3β (Gsk3β) pathway [212]. On the other hand, it has been shown that, after exposure of different concentrations of titanium dioxide NPs (TiO2 NPs) to zebrafish embryos from fertilization to 96 hpf, the hatching time of zebrafish was decreased accompanied by an increase in malformation rate, while no significant increases in mortality relative to controls were observed [213]; moreover, accumulation of TiO2 NPs was found in the brain of zebrafish larvae, resulting in loss of dopaminergic neurons, ROS generation and cell death in hypothalamus. Meanwhile, q-PCR analysis showed that TiO2 NPs exposure increased the *pink1*, *parkin*, *α-syn*, and *uchl1* gene expressions, which are related with the formation of Lewy bodies [213]. Data from zebrafish behavioral phenotype revealed observable effects of silica nanoparticles (SiNPs) on disturbing light/dark preference, dampening exploratory behavior and inhibiting memory capability; furthermore, the relationship between neurotoxic symptom and the transcriptional alteration of autophagy- and parkinsonism-related genes was showed [62]. Similarly, another study showed that 15-nm silica SiNPs produced significant changes in advanced cognitive neurobehavioral patterns (color preference) and caused PD-like behavior compared with 50-nm SiNPs. Analyses at the tissue, cell and molecular levels corroborated the behavioral observations [214]. Both studies demonstrated that nanosilica acted on the retina and dopaminergic neurons to change color preference and to cause PD-like behavior [62,214]. Puerarin has emerged as a promising herb-derived anti-Parkinsonism compound and puerarin nanocristals (PU-NCs) demonstrated no obvious toxic effects on zebrafish, as evidenced by the unaltered morphology, hatching, survival rate, body length, and heart rate; fluorescence resonance energy transfer (FRET) imaging revealed that intact nanocrystals were found in the intestine and brain of adult zebrafish [215,216]. Moreover, other NPs with alleged neuroprotective effects for treating PD, such as polymeric NPs of Ginkgolide B, have shown correct bioavailability and cerebral accumulation in zebrafish models [217].

Similarly, as with PD, some authors have shown neuroprotective and neuroregenerative effects of different NPs in Alzheimer’s disease (AD) zebrafish models. Thereby, it has been shown that casein coated-gold nanoparticles (βCas AuNPs) in systemic circulation translocate across the blood brain barrier (BBB) of zebrafish larvae, sequester intracerebral Aβ42 and its elicited toxicity in a nonspecific chaperone-like manner. This was evidenced by behavioral pathology, ROS, and neuronal dysfunction biomarkers assays, complemented by brain histology and inductively coupled plasma-mass spectroscopy. The capacity of βCas AuNPs in recovering the mobility and cognitive function of adult zebrafish exposed to Aβ was demonstrated [218]. Other study evaluated the role of solid lipid NPs of quercetin (SLN-Q), a flavonoid with multiple pharmacological actions like vascular integrity and regulatory action on the BBB, using pentylenetetrazole (PTZ) induced cognitive impairment of *Danio rerio* species [219]. The intraperitoneal pretreatment of SLN-Q showed an attenuating effect in PTZ induced neurocognitive impairments, along with amelioration of biochemical changes (acetylcholinesterase activity, lipid peroxidation, and reduced glutathione levels), showing differences with fish treated with donepezil. Some authors have demonstrated with confocal image analyses that amphiphilic yellow-emissive carbon dots (Y-CDs) crossed the BBB of five-day old wild-type zebrafish, most probably by passive diffusion due to the amphiphilicity of Y-CDs; furthermore, Y-CDs were internalized by the cells, inhibiting the overexpression of human amyloid precursor protein (APP) and β-amyloid (Aβ) which is a major factor responsible for AD pathology [220]. 

Amyotrophic Lateral Sclerosis (ALS), a fatal neurodegenerative disease affecting the upper and lower motor neurons in the motor cortex and spinal cord, could be ameliorated by reducing the levels of superoxide dismutase I (SOD1). Thus, calcium phosphate lipid coated nanoparticles (CaP-lipid NPs) were developed and tested in zebrafish for the delivery of SOD1 antisense oligonucleotides (ASO) with success, and their preferential accumulation in the brain, blood stream, and spinal cord was observed [221]. 

In addition, the toxic profile of several NPs with potential interest for neurological diseases was also evaluated using zebrafish models. As examples, Carbamazepine or Tacrine were co-administered with PAMAM dendrimers and neurotoxicity, cardiotoxicity, or hepatotoxicity were evaluated in zebrafish larvae [222,223]. These reports provide satisfactory experience in the study of neuro-related disorders using zebrafish models and a good basis for their use as a screening platform to support new nanotechnologies for the treatment of these diseases.

### 4.3. Infectious Diseases

Zebrafish models have been largely used to study infectious diseases, taking advantage of its transparency, possibilities to study both innate and adaptive immunity, and feasibility for the highly controlled administration of pathogens, commonly through microinjection. Prospective treatments are mostly administered by microinjection too, although other routes such as intubation have been described [224]. Most of the studies reviewed in this section have been performed using zebrafish in the first month of life, which does not present a completely developed adaptive immune system but allows for the separate study of macrophages and neutrophils during pathogenic infection [225,226]. Transgenic lines that attach fluorescent proteins to robust macrophages markers, such as *mpeg1* or *csf1ra*, have been used [227]. As an example, Palha et al. infected Tg(*fnφ1*:mCherry) zebrafish larvae with two Chikungunya virus strains, one of which expressed GFP, allowing to follow infection progress and its quantification by flow cytometry. This work, using zebrafish, demonstrated similarities with the process in mammals and a critical role of the IFN response to control the infection. Furthermore, the differential role of macrophages and neutrophils was investigated using a transgenic metronidazole-inducible cell ablation system to deplete macrophages. Neutrophil depletion was studied in *csf3r* knockdown larvae that were highly susceptible to Chikungunya virus, exhibiting a high increase of virus transcripts and mortality [228]. In another study, selective depletion of macrophages by incubation in metronidazole reduced the virulence of infection caused by the *Burkholderia cepacia* complex, revealing the important role of these cells in the infectious process [229]. The easiness for monitoring macrophages in zebrafish allowed extensive studies of tuberculosis and facilitated the observation of granulomas caused by *Mycobacterium marinum* [230,231,232]. Clay et al. took advantage of the transparency of the zebrafish to report a dichotomous role of macrophages in early *M.*
*marinum* infection. First, they performed dual fluorescent antibody detection of L-plastin and myeloperoxidase to confirm that only macrophages and not neutrophils phagocytose bacteria. Most bacteria were found in macrophages (L-plastin positive but MPO-negative). Second, they assessed whether macrophages upregulate inflammatory cytokines as a response to *M. marinum*. They injected separately the bacteria and similar-sized fluorescent beads into the hindbrain, demonstrating that macrophages migrated to the area in response to *M. marinum* but not in response to the beads. Macrophage-defective zebrafish embryos were created by the injection of morpholinos against the *pu.1* gene to evaluate the role of macrophages on mycobacterial growth. Finally, the Tg(*fli1*:EGFP) transgenic line and *pu.1* morphants were used to study bacteria dissemination through the vascular system and the role of macrophages. By injecting red fluorescent bacteria into the bloodstream, they found that control embryos had a higher number of extravascular bacteria than *pu.1* morphants and bacteria injected in the hindbrain could only disseminate out of this space in the control zebrafish with macrophages [231]. Similarly, Davis and Ramakrishnan exploited zebrafish transparency to assess the role of macrophages in tuberculous infection through three-dimensional differential interference contrast microscopy (3D DIC) and fluorescence in vivo microscopy. Embryos were infected with wild type and attenuated *M.*
*marinum*, lacking the ESX-1/RD1 secretion system locus, throughout their experiments. Upon injection of the bacteria into the hindbrain ventricle and daily monitoring, infected macrophages recruited uninfected macrophages in a RD1-dependent manner. Further, the combination of 3D DIC with time-lapse microscopy showed that uninfected macrophages become infected quicker when they are recruited by RD1-competent bacteria. Zebrafish transparency also enabled close observation of the macrophage’s morphology both in WT and RD1 defective bacteria, which proved to be different. Finally, after proving that macrophages become infected when phagocyting dead infected macrophages, granuloma dissemination initiated by macrophages from primary granulomas was also observed. To assess the migration of infected macrophages, they used a bacteria strain that constitutively expresses the Kaede photoactivable protein [232]. Other pathogens relevant for humans (*Candida albicans*, *Herpes simplex*, *Pseudomonas aeruginosa*, *Staphylococcus aureus*, *Streptococcus pneumonia*) [233,234,235,236] and fish (*Vibrio anguillarum*, *Aeromonas salmonicida*, *Yersinia ruckeria*, etc.) [237,238,239] have also been studied using zebrafish models [240,241]. In relation with the study of tuberculosis, Oksanen et al. demonstrated that zebrafish is a useful model for preclinical DNA vaccine development. They vaccinated zebrafish with a combination of plasmids encoding for Ag85B, CFP-10, and ESAT-6, well known mycobacterial antigens, through intramuscular microinjection. Three weeks after immunisation, zebrafish were challenged with a high dose of *M. marinum* (20,500 CFU) by microinjection. Vaccinated fish showed increased survival and the analysis of the bacterial load by qPCR revealed that unvaccinated zebrafish had a higher load than the vaccinated group [242]. Taken together these studies are proof of the versatility of zebrafish to study pathogen infection and dissemination in vivo. Genetic-wise, besides an already existing repertoire of transgenic lines, Clay et al. proved *D. rerio* can be tailored to the needs of the researchers for the study of macrophages by the knockdown of key regulator genes [231].

Several nanotechnology-based-approaches have been developed for vaccination, prophylactic and therapeutic purposes, in the context of infectious diseases, and tested in zebrafish models (Table 3). With prophylactic purposes, Torrealba et al. generated inclusion bodies (IBs) containing TNFα or CCL4 [243]. The resulting nanostructures (IBs) showed good stability under different pH conditions (2.5 and 8). While IB^TNFα^ were cylindrical with a diameter between 380–900 nm and an average length of 1134.6 ± 196.6 nm, the IB^CCL4^ showed spherical shape with a diameter between 220–850 nm. Both IBs were added to the cell culture of ATTC^®^ CRL-2643 (zebrafish liver cells) and RT-HKM (trout macrophages) to evaluate their uptake, showing positive results for both cell types. Further, treatment of RT-HKM with IB^TNFα^ stimulated the expression of pro-inflammatory cytokines. In vivo, zebrafish were immunised with IB^TNFα^, IB^CCL4^, or IB^iRFP-H6^ (an IB containing a control protein) by microinjection and later challenged with *P. aeruginosa*. Zebrafish injected intraperitoneally with both types of IBs exhibited reduced mortality after a challenge with *P. aeruginosa.* Similar inclusion bodies were also used to encapsulate proteins of pancreatic necrosis virus (IPNV), haemorrhagic septicaemia virus (VHSV) and viral nervous necrosis virus (VNNV) for the development of oral prophylactics [224]. IPNV-IBs were barrel shaped and porous with an average width and length of 607 and 734 nm, respectively. VHSV-IBs were round with a width and length of 488 and 608 nm. VNNV-IBs presented an irregular shape with spherical protrusions and a mean diameter of 422 nm. Gene expression analysis of trout macrophages stimulated in vitro with IPNV-IBs and VHSV-IBs showed upregulation of pro-inflammatory markers: *vig1*, *gig2*, *stat1b*, *mx*, *irf7,* and *ccl4*. In vivo uptake of the fluorescently labelled IBs was evaluated by intubating zebrafish adults with the IBs solutions and later analysing gut cells by flow cytometry. Uptake of VHSV-IBs or VNNV-IBs was observed by all fish, while IPNV-IBs were only taken up by 75% of fish. In another study, various recombinant vaccines based on glycoprotein G of viral haemorrhagic septicaemia virus encapsulated in NPs were developed. NPs were produced by complexing poly(I:C) with chitosan, size ranged from 100 to 550 nm with an average diameter of 368 ± 1.3 nm and an average surface charge of +36.2 mV. Fish vaccinated with NPs containing poly(I:C) showed lower mortality rates than non-vaccinated fish or fish vaccinated with NPs without poly(I:C) [244]. Chitosan was also used as a polymer to coat *Piscirickettsia salmonis* membrane nanovesicles for immunisation. The chitosan-coated NPs showed an average diameter of 182.2 ± 4.3 nm and a Z-potential of 31.2 ± 1.8 mV. Immunisation was successful and the upregulation of immune related genes (*IL-8*, *IL-1β*, *IL-10*, and *IL-6*) was reported by analysis of kidney samples [245]. In essence, these articles provide evidence for the use of adult zebrafish as a valid model for immunological studies and as an economic platform for the testing of nano-based approaches designed to improve fish survival, which is of particular interest in aquaculture.

In the search for anti-bacterial compounds to deal with antibiotic resistance, Díez-Martínez et al. assayed the anti-bacterial effect of Auranofin, an FDA-approved drug for the treatment of rheumatoid arthritis, free and encapsulated in nanocapsules of poly(lactic-co-glycolic acid) (PLGA-NPs) in vivo using a zebrafish infection model of *Streptococcus pneumoniae*. PLGA-NPs loaded with Auranofin were spherical with a diameter of 60 nm and a negative surface charge (−30mV). Then, 48 hpf zebrafish embryos were infected by immersion with the pathogen and later treated with free auranofin or auranofin-PLGA-NPs, again by immersion. Encapsulated auranofin was more efficient at rescuing infected embryos in a dose dependent manner than the free drug. Moreover, when compared with encapsulated ampicillin, auranofin PLGA-NPs were still more efficient at increasing survival rates of infected fish [246]. As a cheap, high-throughput model organism, zebrafish could greatly contribute to the initial development of new approaches in the fight against antimicrobial resistance.

Taken together, these findings demonstrate that zebrafish models are very useful for the study of infectious diseases and for the testing of new therapeutic approaches, including NPs. As described above, the evaluation of macrophage and/or neutrophil behavior in this context is of foremost interest, and zebrafish models offer an appropriate environment to study the role of these innate immune cells along the course of the disease and in response to treatment.

## 5. Conclusions

In the last years, the use of zebrafish models in biomedical research has increased substantially to study the cellular and/or molecular basis of human diseases, and for the faster and more economic testing of new compounds, drugs, biomaterials, and nanoparticles. In Figure 3, we can clearly observe the rising number of studies published each year, including zebrafish or nanomaterials, related to toxicity, biodistribution, macrophages, cancer, infectious diseases, or other autoimmune disorders. Manuscripts with a focus on toxicity, macrophages, and cancer are the most frequent, and their number has been consistently increasing in the last decade. Only a few nanomaterials have still been tested in zebrafish models, but the number of this type of studies is also clearly increasing.

In addition to economic and ethical issues, zebrafish models provide excellent opportunities for genetic modifications and for in vivo evaluation/tracking of innate immune cells, such as macrophages. Such unique features make zebrafish amenable to a multitude of methodologies and the establishment of disease models which have already proven viable to study the interaction of macrophages with nanoparticles. These technical advances have been used for a precise toxicological and/or biodistribution testing of nanomaterials with safety or medical purposes. Following a similar trend, we foresee an increase in the testing of nanotechnological approaches for the treatment of cancer, infectious disease, or other autoimmune disorders, using zebrafish models of disease, and providing further information about the role of macrophages in the initiation, progression, and remission of the disease over the course of the treatment. Ultimately, we expect that these studies will contribute to the safe use of nanotechnologies and to their translation towards the clinic, providing new solutions for patients. 

## Figures and Tables

**Figure 1 nanomaterials-11-01784-f001:**
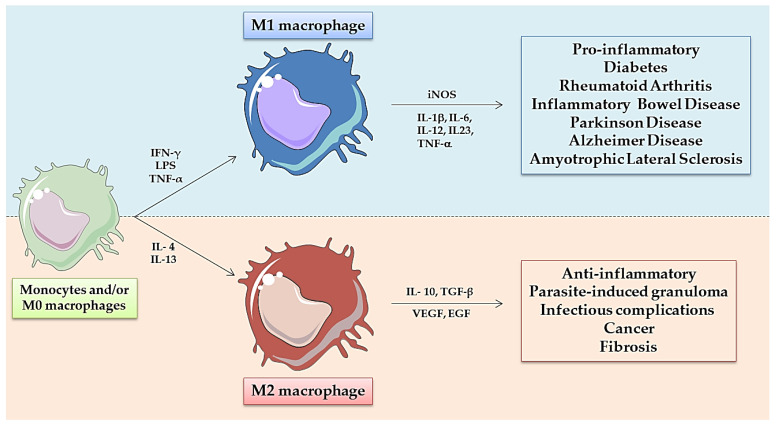
Macrophages originate from monocytes or tissue-resident macrophages. In response to different microenvironmental stimuli macrophages polarize towards an M1-like or M2-like phenotype, and the excessive accumulation of macrophages with a particular phenotype has been correlated with a poor prognosis in some diseases (on the right). In pathological tissues, these macrophages frequently contribute to the development and progression of the disease, thus their reprogramming towards an opposite polarization status has been recognized as an important therapeutic strategy.

**Figure 2 nanomaterials-11-01784-f002:**
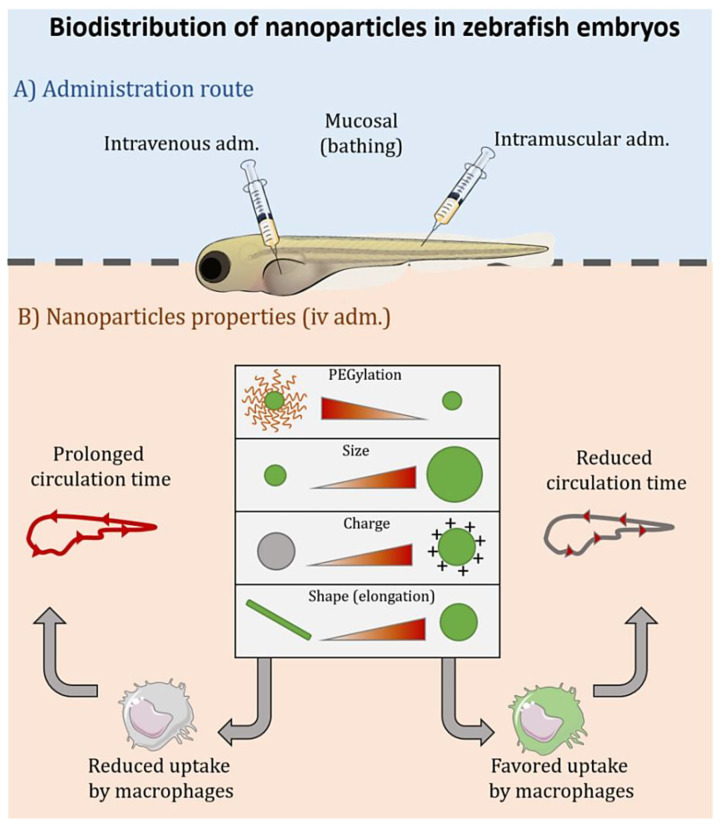
Biodistribution of nanoparticles in zebrafish is related to their interaction with macrophages. The biodistribution of nanoparticles in zebrafish embryos is mainly dictated by (**A**) the administration route and (**B**) the physicochemical properties of the nanoparticles. Among the characteristics that most impact the circulation time after intravenous administration are the degree of PEGylation, the particle size, the surface charge and the shape of the particles. By tuning these properties, the uptake by macrophages can be minimized and the circulation time can be prolonged.

**Figure 3 nanomaterials-11-01784-f003:**
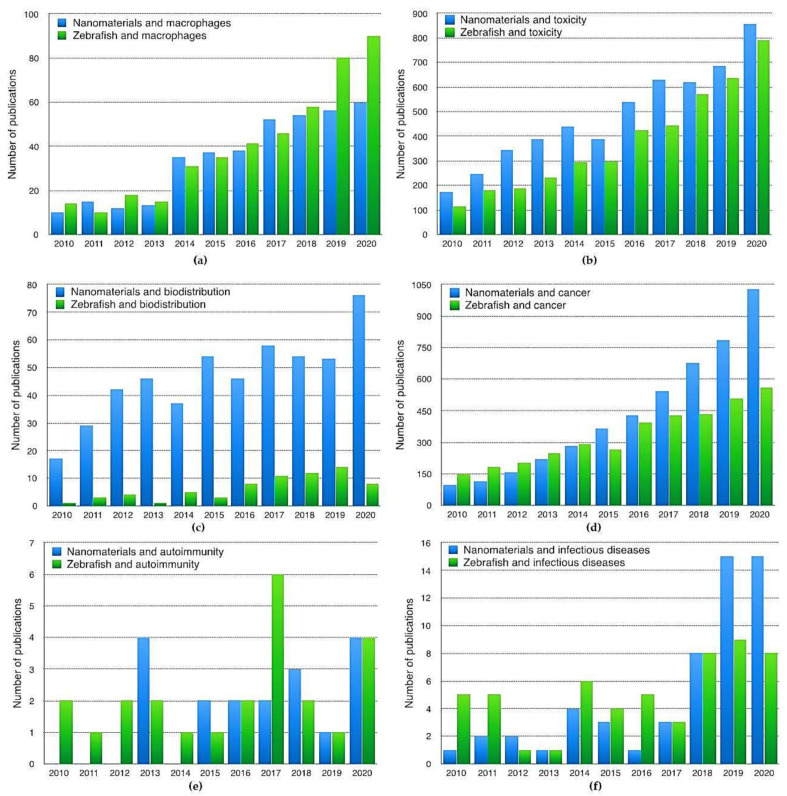
Number of publications each year in Pubmed for indicated terms from 2010 to 2020: (**a**) Nanomaterials and macrophages/Zebrafish and macrophages; (**b**) Nanomaterials and toxicity/Zebrafish and toxicity; (**c**) Nanomaterials and biodistributrion/Zebrafish and biodistribution; (**d**) Nanomaterials and cancer/Zebrafish and cancer; (**e**) Nanomaterials and autoimmunity/Zebrafish and autoimmunity; (**f**) Nanomaterials and infectious diseases/Zebrafish and infectious diseases.

**Table 1 nanomaterials-11-01784-t001:** Evaluation of nanoparticle-based approaches for cancer treatment using xenograft zebrafish models.

Nanosystem	Drug	Mechanism	Tumor Type/ Cell Line	Zebrafish Stage	Injection Site	NPs Delivery	Remarkable Results	Reference
PtPP-HA	Kiteplatin pyrophosphate	Apoptosis through DNA platination	Breast cancer/ MDA-MB-231-GFP	48 hpf	Duct of Cuvier	Co-injection with cells	Decrease in breast cancer cells survival	[129]
Zinc oxide NPs	-	Apoptosis and ROS induction	Gingival squamous cell carcinoma/ Ca9-22-DiL	48 hpf	Yolk sac	Immersion/ 48 hpf	Dose-dependent antitumoral activity	[130]
PMOsPOR-NH_2_/TPE-PDT	-	Porphyrin photosensitivity and ROS production	Breast Cancer/ MDA-MB-231-GFP	30 hpf	Duct of Cuvier	Pre-treatment of cancer cells	Complete extinction of cancer cells	[131]
PORBSNs/ TPE-PDT	-	Porphyrin photosensitivity and ROS production	Breast cancer/ MDA-MB-231-GFP	24-30 hpf	Perivitelline space	Intravenously/ 4 dpi	Decrease of the tumor area	[132]
PAMAM-GC/DOX/γ-radiation	DOX	GC radiosensitivity increases DOX release	Uterine cervical carcinoma/ HeLa- CSFE	48 hpf	Yolk sac	Immersion/ 1 dpi	Synergistic antitumoral effect for the combination of GC/DOX and radiotherapy	[133]
PAMAM-DOX-siHIF	DOX/siHIF	NPs responsiveness to hypoxia and increased drug release	Breast cancer/ MCF-7-CM-DiL	48 hpf	Perivitelline space	Intracardiac injection/1 dpi	Feasibility of the cooperative strategy for in vivo applications	[134]
Nanogel^DOX^	DOX	Hydrazone sensitivity to pH accelerate drug release	Melanoma/ B6-RFP or GPF	48 hpf	Neural tube	Intravenously/ 1 dpi	Selective accumulation of the NPs in the tumor and reduction in tumor growth	[135]
Tf-DOX-ReSi-Au	DOX	Enhanced tumor targeting by interaction between Tf and Tf receptor	Colorectal cancer/ HCT116-GFP	48 hpf	Yolk sac	Retro-orbital injection/1 dpi	Antitumoral activity without DOX-related cardiotoxic effects	[136]
PEG liposomes	-	-	Melanoma/ Melmet 5-dsRed Kidney/HEK293-mCherry	48 hpf	Duct of Cuvier	Intravenously/ 2 dpi	NPs accumulation in human tumor structure, low macrophage uptake and high survival rate	[72]
PEG-PDPA-DOX	DOX	Polymersomes release the drug only at low pH	Melanoma/ B6-RFP or GPF	72 hpf	Neural tube	Intravenously/ 1 dpi	Selective accumulation of NPs in the tumor area, increased cancer cell apoptosis and reduced proliferation	[137]

dpi: days post-injection; DOX: doxorubicin; PAMAM-DOX-siHIF: G4.5 polyamidoamine dendrimers loaded with DOX and hypoxia-inducible factor 1a siRNA; PAMAM-GC/DOX: G4.5 polyamidoamine dendrimers with l-cysteine and loaded with DOX; PEG-PDPA-DOX: Poly(ethylene glycol)-block-poly(2-(diisopropyl amino) ethyl methacrylate) NPs loaded with DOX; PMOsPOR-NH2: porous porphyrin-based organosilica NPs; PORBSNs: non-porous porphyrin-based bridged silsesquioxane NPs; PtPP-HA: kiteplatin-pyrophosphate-loaded hydroxyapatite NPs; Tf-DOX-ReSi-Au: Redox-responsive silica-gold nanocomposites functionalized with transferrin and loaded with DOX; TPE-PDT: two-photon-excited photodynamic therapy.

**Table 2 nanomaterials-11-01784-t002:** Summary of nanoparticles tested in zebrafish models of autoimmune diseases: inflammatory bowel disease, type I diabetes mellitus, parkinson’s disease, alzheimer’s disease, amyotrophic lateral sclerosis.

Disease	Disease Induction	Zebrafish Stage	Nanosystem	NPs Delivery	Remarkable Results	Reference
Inflammatory Bowel Disease	Copper NPs- induced intestinal developmental defects	From 0 hpf	Copper NPs	Immersion	CuNPs cause intestinal developmental defects via inducing ER stress and ROS generation, which corresponds with elevated serum copper levels in IBD patients	[174]
Type 1 diabetes mellitus	Glucose-induced diabetic zebrafish	Adult	EP/AgNPs	Immersion	Hyperglycemia amelioration	[189]
Type 1 diabetes mellitus	STZ- induced diabetic retinopathy	Adult	Quercetin NPs	Intraperitoneal injection	Reduction of glycemia and improvement of morphological, behavioral and biochemical parameters linked to retinopathy	[190]
Type 1 diabetes mellitus	-	From 4 hpf	pMHC-NPs	Immersion	Neither off-target toxicity, nor morphological abnormalities	[191]
Parkinson’s Disease	MPTP-induced parkinsonism	Adult	BmE-PtNPs	Intraperitoneal injection	Significant reversion of toxic effects of MPTP by increasing the levels of dopamine, GSH, GPx, catalase, SOD and complex I, and reducing levels of MDA	[211]
Parkinson’s Disease	MPTP-induced parkinsonism	From 72 hpf	Schisantherin nanocrystals	Immersion	Reversed dopaminergic neuronal loss and locomotion deficiency by the activation of the Akt/Gsk3β pathway	[212]
Parkinson’s Disease	-	From 0 hpf	Titanium dioxide NPs	Immersion	Loss of dopaminergic neurons, ROS generation and cell death in hypothalamus. Increased Lewy bodies-related markers.	[213]
Parkinson’s Disease	-	Adult	Silica NPs	Immersion	Changes in dopaminergic neurons with disturbed light/dark preference, dampened exploratory behavior, inhibited memory capability and PD-like behavior	[62,214]
Parkinson’s Disease	-	From 6 hpf	Puerarin Nanocristals	Immersion	Promising anti-Parkinsonism NCs. Unaltered morphology, hatching, survival rate, body length and heart rate	[215,216]
Parkinson’s Disease	-	From 120 hpf	Ginkgolide B-PEG-PCL NPs	Immersion	Correct bioavailability and cerebral accumulation in zebrafish models	[217]
Alzheimer’s Disease	Aβ- induced toxicity	Adult	Casein coated-gold NPs	Retro-orbital injection	Inhibition of Aβ toxicity and recover of the mobility and cognitive function	[218]
Alzheimer’s Disease	PTZ- induced cognitive impairment	Adult	Solid lipid NPs of Quercetin	Intraperitoneal injection	Attenuation of PTZ-induced neurocognitive impairments and amelioration of biochemical changes	[219]
Amyotrophic Lateral Sclerosis	-	From 96 hpf	ASO- CaP-lipid NPs	Brain, spinal cord, intravenous and retro-orbital injection	Successful delivery and preferential accumulation in brain, bloodstream and spinal cord	[221]

APP: Human amyloid precursor protein; ASO-CaP-lipid NPs: SOD1 antisense oligonucleotide-calcium phosphate lipid coated NPs; Aβ: β-amyloid; BmE-PtNPs: Bacopa monnieri platinum NPs; EP/AgNPs: *Eysenhardtia polystachya*-loaded silver NPs; ER: Endoplasmic reticulum; GSH: Glutathione; GSH-Px: Glutathione peroxidase; MDA: Malondialdehyde; MPTP: 1-methyl-4-phenyl-1,2,3,6-tetrahydropyridine; STZ: Streptozotocin; PEG-PCL NPs: Poly(ethylene glycol)-co-poly(ε-caprolactone); pMHC-NPs: Peptide-major histocompatibility complexes NPs; PTZ: Pentylenetetrazole; SOD: Superoxide dismutase.

**Table 3 nanomaterials-11-01784-t003:** Summary of nanoparticles tested in zebrafish models of infectious diseases.

Disease Induction	Zebrafish Stage	Nanosystem	NPs Delivery	Immune Cells Behavior	Remarkable Results	Reference
*Pseudomonas aeruginosa* infection	Adult	Nanostructured cytokines (IB^TNFα^, IB^CCL4^)	Intraperitoneal injection	Interaction of IBs with immune cells	Prophylactic potential in vivo	[243]
-	Adult	IPNV, VHSV and VNNV-encapsulated IBs	Oral intubation	-	Successful NPs uptake in gut cells after oral administration	[224]
VHSV infection	Adult	Viral glycoprotein G encapsulated in chitosan-poly(I:C) NPs	Intraperitoneal injection	Upregulation of antiviral cytokines	Significant protection against VHSV through induction of anti-viral state	[244]
*Piscirickettsia salmonis* infection	Adult	Chitosan-coated MVs from *P. salmonis*	Intraperitoneal injection	Upregulation of immune related genes	Successful immunisation. Potential use of chitosan-coated MVs for vaccination	[245]
*Streptococcus pneumonia* infection	From 48 hpf	Auranofin-PLGA-NPs	Immersion	-	Auranofin-NPs capability of decreasing the bacterial population compared to free drug	[246]

IBs: Inclusion bodies; IPNV: Pancreatic necrosis virus; VHSV: Viral hemorrhagic septicemia virus; VNNV: Viral nervous necrosis virus; MVs: Membrane nanovesicles; PLGA-NPs: Poly(lactic-co-glycolic acid) NPs; LPS: Lipopolysaccharide.

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
