# Peer review of "Zebrafish Models for the Safety and Therapeutic Testing of Nanoparticles with a Focus on Macrophages"

_nanomaterials, 2021, doi:10.3390/nano11071784_

Round 1

Reviewer 1 Report

It is a nicely written review. I would only recommend to add a couple of references, which were amused by the authors. One demonstrates a highly challenging injection of nanoparticles in zebrafish embryo: J. Vis. Exp,  (159), e61187, 2020. This work shows how to do it in a user-friendly video format. The second is the use of highly fluorescent and biocompatible nanoparticles prepared from cellulose acetate: Materials Today, v.23, pp.16-25, 2019.

Author Response

Response to Reviewer 1

Point 1: It is a nicely written review. I would only recommend to add a couple of references, which were amused by the authors. One demonstrates a highly challenging injection of nanoparticles in zebrafish embryo: J. Vis. Exp,  (159), e61187, 2020. This work shows how to do it in a user-friendly video format. The second is the use of highly fluorescent and biocompatible nanoparticles prepared from cellulose acetate: Materials Today, v.23, pp.16-25, 2019.

 Response 1: As suggested by the reviewer, we have added the indicated references as follows:

Lines 151-152:

“The usual methods of exposure to new compounds or nanomaterials are microinjection and immersion [44, 45].”

Lines 440-443:

“Also with targeting purposes, Peng et al., synthesized fluorescent cellulose acetate NPs functionalized with folate groups for their preferential accumulation in epithelial cancer cells overexpressing folic acid receptors [140]. These NPs can be tuned within the entire UV–VIS-NIR spectrum and were capable to target tumors in vivo.”

Reviewer 2 Report

Comment

The manuscript by López et al, reviewed the studies using zebrafish for toxicological and bio-distribution testing of nanoparticles, especially focus on macrophage-nanoparticle interaction in various diseases, including cancer and autoimmune, neuroinflammatory, and infectious diseases. The authors have performed a comprehensive review of the safety and therapeutic testing of nanoparticles with a focus on the zebrafish animal model of macrophages. The manuscript is well written and easy to understand. However, before the manuscript can be judged suitable for publication, there are two minor issues that can be improved:

  1. In the 3.1. Toxicological studies section (after Line 175), although the authors have reviewed inorganic-based NPs and/or organic-based nanomaterials induced toxicity (for example, developmental or neuron toxicity) in the zebrafish model, I suggest the authors can add the latest toxic effect of nanomaterials in the zebrafish innate immune system, such as AgNPs and CuO NPs, etc. Because the review article is focused on zebrafish macrophages (one of the innate immune cells), which could help the reader figure out the new information in the toxicological studies section.
  2. In table 1, some of the abbreviations did not give the full name in the notes to the table, for example, DOX. The authors should recheck the table.

Author Response

Response to Reviewer 2

Point 1: The manuscript by López et al, reviewed the studies using zebrafish for toxicological and bio-distribution testing of nanoparticles, especially focus on macrophage-nanoparticle interaction in various diseases, including cancer and autoimmune, neuroinflammatory, and infectious diseases. The authors have performed a comprehensive review of the safety and therapeutic testing of nanoparticles with a focus on the zebrafish animal model of macrophages. The manuscript is well written and easy to understand. 

Response 1: We appreciate the positive feedback from the reviewer.

Point 2: In the 3.1. Toxicological studies section (after Line 175), although the authors have reviewed inorganic-based NPs and/or organic-based nanomaterials induced toxicity (for example, developmental or neuron toxicity) in the zebrafish model, I suggest the authors can add the latest toxic effect of nanomaterials in the zebrafish innate immune system, such as AgNPs and CuO NPs, etc. Because the review article is focused on zebrafish macrophages (one of the innate immune cells), which could help the reader figure out the new information in the toxicological studies section.

Response 2: In the section “3.1. Toxicological studies”, we have added a couple of articles on immunotoxicity of metallic nanoparticles as follows:

Line 213-216:

“Similarly, it has been reported that several types of metallic NPs, such as AuNPs, AgNPs or ZnONPs may induce oxidative stress and disrupt signaling pathways related to innate immune responses [78-80]”.

Point 3: In table 1, some of the abbreviations did not give the full name in the notes to the table, for example, DOX. The authors should recheck the table.

Response 3: We have reviewed the list of abbreviations, and we have added DOX in the notes of the table and AuNPs and AgNPs in the whole list of abbreviations.
